# Getting the FACS: A Protocol for Developing a Survey Instrument to Measure Carer and Family Engagement with Mental Health Services

**DOI:** 10.3390/ijerph192316279

**Published:** 2022-12-05

**Authors:** Darryl Maybery, Andrea Reupert, Irene Casey Jaffe, Rose Cuff, Zoe Duncan, Addy Dunkley-Smith, Anne Grant, Melissa Kennelly, Bjørg Eva Skogøy, Bente Weimand, Torleif Ruud

**Affiliations:** 1Department of Rural Health & Indigenous Health, Monash University, Warragul 3820, Australia; 2School of Educational Psychology & Counselling, Monash University, Melbourne 3800, Australia; 3Harvard T.H. Chan School of Public Health, Harvard University, Boston, MA 02115, USA; 4Satellite Foundation, Melbourne Central, Melbourne 3000, Australia; 5School of Nursing and Midwifery, Queen’s University Belfast, 97 Lisburn Road, Belfast BT9 7BL, UK; 6Department of Rural Health & Indigenous Health, Monash University, FaPMI Strategy, Mildura 3500, Australia; 7Nordland Research Institute, 8049 Bodø, Norway; 8Center for Mental Health and Substance Abuse, University of South-Eastern Norway, 3004 Drammen, Norway; 9Division Mental Health Services, Akershus University Hospital, 1478 Lørenskog, Norway; 10Institute of Clinical Medicine, University of Oslo, 0450 Oslo, Norway

**Keywords:** parental mental and physical illness and disability, young carers, youth adjustment to parental illness, family health

## Abstract

Government policies recommend, and all stakeholders benefit, when mental health services meaningfully engage with carers and family. However, health service engagement with carers is inadequate, and often non-existent with children whose parents are service users. There are seven fundamental ways that carers and families want to be integrated with and engaged by health services but current survey instruments do not capture these seven engagement practices. This protocol describes the development of two closely aligned Family and Carer Surveys (FACS) to measure engagement of service users in mental health services. The new measures are based on the seven engagement themes and a conceptual distinction between the carer and family, with particular focus on where the service user is a parent. The instruments will be developed in five stages; (1) item generation (2) Cognitive pretesting of survey (3) preliminary item content quantitative assessment (4) psychometric analysis of a large data collection and (5) selection of items for short form instruments. These steps will operationalise the seven fundamental ways that families and carers want to be engaged with mental health services, thereby providing valid and reliable measures for use in research and benchmarking of carer and family engagement.

## 1. Introduction

Both mental health service users [1,2,3,4] and their carers, family and friends benefit when carers and family are engaged by health services [5,6,7]. Benefits from integrated care are found across health conditions such as cancer [8], mental illness [6], when addressing carers’ social and emotional needs [9] and in families where parents who have a mental illness illness [10]. The importance of engaging with carers and family is also noted in many western government policy and procedures, for example, in the UK [11] the USA [12], New Zealand and Australia [13], Canada [14], Norway [15] and worldwide [16].

A carer is defined as “…someone who is actively supporting, assisting or providing unpaid care to…someone…who has received, is receiving, or is seeking, treatment and support from…health services.” [17] (pp. 1–2). The term carer has been criticised for its connotation of dependency and lack of acknowledgement of carer and service user reciprocity [18] and although many family members undertake caring responsibilities “some family members…will identify more so with the characteristic of their relationship, for example, parent, child, partner, sibling” [17] rather than as ‘carers’ [19]. The term carer appears best applied from the nomenclature used by respective family member or friend to describe their relationship to the service user [19].

The term ‘family’ “…might include biological relatives, intimate partners, ex-partners, people in co-habitation, children, friends, those with kinship responsibilities…” [20]. Family may involve either one’s family of origin and/or family of procreation [21]. In this paper we focus on where the service user is a parent with dependent children. One in five children grow up with at least one parent with a mental health problem [20] and a large percentage of adults attending state mental health services are also parents [22,23]. These children are at high risk of being taken into care [24], of school failure and dropout [25] and of acquiring a mental health condition themselves [26] often with their problems continuing into adulthood [27].

Prevention and early intervention benefits can be derived in families where parents have a mental illness, when health services engage with service users about parenting support and respond to the needs of children. In some families, children may identify as, and play the role of carer, for their siblings and parent/s. Sometimes considered a distinct and often at risk group in society, the term refers young people under 18 years who provide regular and ongoing support to a family member with a major health problem [28]. While not all young people with an ill family member are young carers [28] research has shown that some young carers have negative experiences with health services and/or experience barriers when seeking support for their own health needs [29].

Various studies demonstrate the value of health services engaging with carers and family members [30] including the promotion of adaptive coping [31]. However, Peters and colleagues highlighted problems that many carers experience with health services, for example “…carers who reported more problems with health and social services had worse quality of life and higher strain” [32] (p. 1). Peters et al., also concluded by stressing the importance of the health care sector “…appropriately supporting carers…” [32] (p. 8). Overall, family and carer engagement by services is at best modest [33] and generally “…little information is available on the impact of support of health and social care services on caregiver well-being” [33]. Compounding the problem is that, to the best of our knowledge, there are no valid or reliable measures of the optimal ways that health care services might engage with family and carers. This gap limits the ability of health services to measure engagement practices for quality assurance, evaluation, and/or research purposes [30].

As well as being confirmed empirically by carers and family [20] the seven practices were considered fundamental based on evidence from previous systematic reviews [34], research conducted with clinicians [35], service users [36] family members [37] and structures within Northern Irish and Australian practice audit tools [20]. The evidence from audit tools was considered important, as they are distinct from surveys, being used to audit patient files recorded during their attendance at mental health services. They provide strong content validation (from an alternate source to surveys/literature) of the 7 fundamental practices. The fundamental practices are to:Identify and acknowledge family and carers;Engage and communicate with family and carers;Involve family and carers in planning/collaboration in service user’s treatment;Assess vulnerable family member or carer’s needs;Provide or offer ongoing support to family and carers;Provide psycho-education to family and carers, andProvide or recommend referrals for family and carers.

Although the practices are fundamentally important, they “…should not replicate existing services but supplement, extend and support services. Their place as equal but different to what services can provide to service users…” [20] (p. 8). The ‘fundamentals’ are aligned with the most recent policy update of the National Institute for Health and Care Excellence guidelines that clearly indicate that health services should offer information to carers about the service users condition and management, provide shared communication, education and collaboration—including carers in decision making, along with making an assessment of the carers own needs [11].

From a conceptual perspective, more than one single theoretical approach is needed to acknowledge the seven practices. From the mental health worker perspective, Reupert and colleagues highlight family focused practice as a potential framework [21]. From the carer and family perspective, they summarise multiple conceptual models such as Brofenbrener’s ecological framework, Goodman and Gotlib’s integrative model of risk and Falkov’s family model amongst others. They highlight that some theories provide a broad ecological view of family and carers in relation to their social and community circumstances and other approaches highlight specific disorders (e.g., anxiety) and how they impact upon relationships. They also illustrate that these different theoretical foci have been drawn from different stakeholder experiences including clinical, research and family and carer experiences. Of note here is that the seven practices have been appraised and rated by carer and family stakeholders as fundamental practices.

Recently, Lin et al., undertook a comprehensive review of 32 carer survey instruments to determine “…how caregivers interact with larger social systems and the impacts of factors such as financial strain, lost time from leisure activities, and the availability of health and social services” [38] (p. 615). They examined items and subscales that measure aspects of “caregiver work demands, resource needs, resource utilisation and costs” of caring [38] (p. 614). Although not specifically focusing on how services might engage with carers and family, the review provides an opportunity to examine those instruments for relevance to the aims of this protocol. Ten of the 32 instruments were considered to potentially measure some, or all of the seven fundamental engagement practices. Table 1 summarises and provides commentary regarding which engagement practices that are potentially quantified in each of those carer instruments.

Table 1 shows that the carer measures currently available focus on multiple aspects of carers and families experiences including: carers’ experiences [41]; needs [39,45]; a range of carer wellbeing factors (e.g., burden [42,43] outcomes including quality of life [44]; resource requirements [32,40], and access satisfaction [46]. Seven of the instruments included items classified as meeting two to four of the seven engagement fundamentals. None had items reflecting all seven practices. Of notable mention was the Australian developed, Carer Experience Survey [47]. Many of the 27 CES quantitative items focus on the needs of the service user although some focus upon carer wellbeing (e.g., [rate] Your overall wellbeing). Of relevance there is only one item on the CES focuses on the carer or family member’s needs (i.e., item *19. Information about carer support services (such as local groups, carer consultants, counsellors)* and notably absent are items regarding such things as receiving psychoeducation and items pertaining to children and young carers [47]. Such items would seem to be very important inclusions in measures of carer and family engagement with services.

In sum, Government policy, published research and relevant grey literature (i.e., regarding the CES) highlight the fundamental ways that health services should engage with family and carers. An examination of previous instruments used to measure carer and family relationships with health services indicates that no measure fully measures these fundamental domains. This protocol outlines a methodology to develop a valid and reliable survey instrument that measures the seven fundamental ways that services can engage with carers and families, from the perspective of both those who identify as carers, and also from the perspective of other family members particularly those families where a parent has a mental illness. This protocol outlines research that will develop closely related Family (where parent has a mental illness) and Carer Survey (FACS) instruments that are valid and reliable. They will capture the seven fundamental ways that families and carers want to be engaged with by mental health services.

## 2. Materials and Methods

The conceptual structure of the new measure will be based on the seven fundamental practices highlighted by Maybery and colleagues [20]. Table 2 shows these seven core practices and the scale items that will be developed. There will be two different but closely related measures, namely one for where the service user is a parent and the other for when the service user is not a parent. This separation includes the conceptual distinction of being a carer or family member for service users who have parental responsibilities and those who do not. A growing literature on parental mental illness illustrates the critical need for mental health services to respond to the service users parenting responsibilities and to engage with carers and the service user about the welfare of the service users’ children [22,23,48,49]. In many countries, there is now a legal (e.g., Norway [15]) and/or ethical responsibility for health services to identify and respond to the needs of service users’ children.

Two distinct but closely related questionnaires will be developed with items reflecting the seven fundamental practices (see example items under carer and family column headings in Table 2). The figure shows example items according to carers where the service user is not a parent and items for family where the service user is a parent (both parent and child items represented).

Building on the theoretical and conceptual structure outlined above, the survey instruments will be developed employing five stages; (1) item generation and International panel analysis of content, (2) Cognitive pretesting of survey, (3) preliminary quantitative (i.e., item reliability/single factor congeneric models) assessment of item content (4) psychometric analysis (component structure, item-scale correlations, construct validity) of longer form questionnaires, and (5) selection of items for development of short form instruments (as per [50,51,52]). Figure 1 illustrates the stages of development of the FACS.

### 2.1. Item Generation and International Panel Analysis of Content

An initial item pool of approximately 50 items each for the carer (non parent) and parent instruments will be developed by the first author, based upon 40 interviews and over 300 text based verbatim comments from a previous study with carers and family including adult children of parents with mental health problems [20]. A reference group of International experts will critique, modify and add items resulting in approximately seven items per construct for each measure. These experts from Australia, Canada, Ireland, Norway and the USA, come from the fields of psychology, nursing and psychiatry, are part of a collaborative that meets every two years to undertake research in the field of parental mental illness [53]. Over half of group have lived experience of mental health issues including being parents, children and/or carers. The Prato Collaborative for Change in Parent and Child Mental Health aims to promote workforce change in relation to children who have parents with a mental illness.

### 2.2. Cognitive Pretesting of Survey

The items and instrument instructions will then be pretested by carers, family service users and clinicians (*n* = 12–15). Participants will complete the instruments and then in interview provide critique and suggest modifications to the items. This will be undertaken for both carer and parent measures. The semi-structured interviews will be undertaken by trained interviewers and audio-taped and transcribed. Participants will be paid for their time.

### 2.3. Preliminary Quantitative (i.e., Item Reliability/Single Factor Congeneric Models) Assessment of Item Content

The initial version of the instruments will be distributed to approximately 30 carers and 30 children of parents with a mental illness, sampled by convenience using social media (e.g., Facebook), targeted sampling to carer and parental mental illness groups along with snowball/respondent driven sampling. One factor congeneric analysis will be undertaken on each of the seven domains to examine for loading, reliability and for the removal of inadequate loading items. At this point the items will be reduced to four items for each engagement concept for carers (total 28 items) and one parent and one child focused item for where a parent has the mental health concern. Financial assistance may be provided to participants for the time spent completing the measures.

### 2.4. Psychometric Analysis of Long Form Questionnaires

Psychometric analyses will be undertaken, employing exploratory (to explore the statistical structure) and confirmatory (attempting to confirm the seven fundamental engagement practices) analyses of the final version of the measures. Data from approximately 250 carers and 250 adult children of parents with a mental illness participant, sampled as indicated above. A total of 500 participants meets the sampling requirements for these structural statistical analyses including assessments of reliability and validity [54] (see Flora and Jessica K. Flake 2017 who detail 10 participants per scale item (i.e., 42 = 420 participants) and that 500 or more participants is a very good sample size for such analyses). Along with the target instrument, participants will complete measures of mental health and wellbeing, carer burden and the carer experiences of service instruments. Initially analyses will focus upon confirming item-component structure and subscale reliability. This will be followed by correlation analyses between the seven subscales of the target instrument and additional measures to examine the construct and concurrent validity of the former. Financial assistance may be provided to participants for the time spent completing the measures.

### 2.5. Short Form Instruments

Using the above data, further analyses will be undertaken to determine the best items for inclusion on the short form of the measures. Three criteria will be used to choose items; first, for their stronger correlation with other related constructs (e.g., carer burden); second for their item-total reliability scores, and; third for their strength of component loading/reliability as part of the short form measure. This inclusion criteria will ensure that the most reliable items with the strongest correlations are chosen from the long form that then contribute the strongest loadings to the short form structure and measurement characteristics.

In terms of the preliminary quantitative phase, distributions of item and subscale responses will initially be examined for presence of skewness, kurtosis and possible ceiling and floor effects. The items within subscales will then be examined using Cronbach alpha and single factor congeneric models to assess item content and removal of inadequate loading items. Analysis of the long form questionnaires will employ principal component analyses of the complete measures to examine component structure, correlational analyses of subscales with additional instruments to determine concurrent and construct validity and reliability will be examined via Cronbach’s alpha. Additional structural analyses including correlations with related constructs, item total reliabilities and component loadings to determine the items to be included in the short form instruments.

## 3. Discussion

This protocol presents the theoretical background, conceptual structure and qualitative and quantitative methods to be employed in the development of surveys that measure how health services engage with carers and families. The series of steps in development of the FACS instruments will operationalise the seven fundamental ways that families and carers want to be engaged with health services. This process will provide new valid and reliable carer and parent instruments that reflect the seven engagement concepts determined from previous reviews [34,35] and empirical work that has confirmed the seven fundamental ways that families and carers want to be engaged with health services [20,55].

If successfully developed, the survey instrument will allow for the future critique and testing of the seven engagement concepts but more importantly to give policy makers and health services the opportunity to assess, quantify and benchmark their carer and family engagement policies, procedures and strategies. The FACS could be used for assessing service delivery and quality improvement and could give direction to workforce development, training and evaluation. The new measure also has potential to develop a program of research about if and what aspects of carer and family engagement contributes to outcomes for; service users (e.g., medication compliance), carers (e.g., carer burden) and families (e.g., benefits to parenting and children). At a more practical level the FACS has the potential to enhance the inclusion and empowerment of family and carers in relation to their own and service user needs.

There are several limitations with the design of the study outlined. Recruitment of the study sample is partially via social media, which has a self-selection bias and potentially not be representative of the carer and family populations including different subgroups especially older relatives [44]. Additionally, the chronicity and severity of the service users’ illness is not considered and will require future research in wider populations of carers and families [56]. Data analysis and findings will be compromised by a single sample of participants including the lack of test–retest reliability analysis due an inability to follow participants up due to their anonymous involvement in the study. Ref. [57] requiring later additional studies to confirm the psychometrics of the current research.

## 4. Conclusions

Building upon previous research, this protocol describes the next important step in the developing measures of how mental health services should engage with carer, family, and parents. Once developed the instruments have the potential to improve stakeholder wellbeing through better understanding how key carer and family stakeholders should be integrated with service user treatment [58,59,60].

## Figures and Tables

**Figure 1 ijerph-19-16279-f001:**

Protocol stages from item generation to short form analysis of FACS long and short form instruments.

**Table 1 ijerph-19-16279-t001:** Item content of 10 carer/family measures (as identified by [38]) as applied to the seven * fundamental engagement practices (√ = one item, √√ = more than one item identified from scale).

Carer/Family Scale	1	2	3	4	5	6	7	Comment
Carer experiences of health and social care [32] 12 items.		√	√	√√	√√			Most items focused on the amount of resources received from health and social care services. Social care services includes organisations designed to provide support to carers and families.
Carers’ needs assessment for schizophrenia [39] 18 items.		√	√			√√	√√	This measure is an assessment of the problems and required interventions for carers/families. This is not a measure which evaluates the way a service engaged with the carers/family, although some items may address this issue.
Inventory of Mental Health Resources [40] 7 items.					√			Asks caregiver if they accessed resources to support their own needs or service user needs (e.g., crisis resources such as emergency rooms, community crisis centre), health services, child day care, psychological services and support services (e.g., peer support groups, on-line services for consumers). Not a measure of how services engaged with carer/family.
Experience of care-giving inventory [41] 66 items.	√√	√√				√√	√	8 relevant items focused upon “Problems with services”.
Mood Disorder Burden Index [42] 32 items.		√		√				Most items focused on service users’ issue/concerns with health service. Two items related to caregiver health.
Caregiver Reaction Assessment [43] 27 items.				√				This instrument is frequently used to measure carer burden. The only items perhaps associated with services engagement is “Difficult to get help.” and “Feel abandoned” Although this Item is within the “Lack of family support” Factor—and may therefore refer to difficulty getting help from other family members as opposed to services.
Schizophrenia Caregiver Quality of Life questionnaire [44] 25 items.		√	√		√			Measure focuses on carer quality of life.
Contacts with health/social services with re needs [45] 16 items.					√		√	Based on a measure by Peto and colleagues, most items focused on the amount of resources received from health and social care services (e.g., In-home respite care, Social worker, Alternative medicine).
Modified Opinion Questionnaire on Outpatient Services [46] 21 items.	√	√√				√√		Unable to locate scale that was adapted from satisfaction with access, use, and continuity of mental health services for service users.
Mental Health Carer Experience Survey [47] 37 items, including demographics.	√	√√	√√		√		√√	The survey highlights items focused upon assessment of carer vulnerability (e.g., hopefulness for your future, overall wellbeing) but it does not ask if the health service assesses these issues.

* 1. Identify and acknowledge family/carers; 2. Engage and communicate with family/carers; 3. Involve family/carers in planning/collaboration in consumer’s treatment; 4. Assess vulnerable family member or carer’s needs; 5. Provide or offer ongoing support to family/carers; 6. Provide psychoeducation to family/carers, and 7. Provide or recommend referrals for family/carers.

**Table 2 ijerph-19-16279-t002:** Engagement practice domains and items where the service user is and is not a parent.

7 Engagement Domains	Carer: Service User Is Not Parent	Family: Service User Is Parent	
	Carer Items	Parent Items	Child Items
1. Identify and acknowledge family and carers	*The health service checked on how well I was coping as a carer or family member*	*The health service knew the service user was a parent.*	*The staff of the health service knew the service user had child(ren).*
2. Engage and communicate with family and carers	*The health service respected me as a family and/or carer*	*The health service talked to the service user about their parenting*	*The health service talked with the service user’s child(ren)*
3. Involve family and carers in service user planning/collaboration regarding treatment	*The health service made time for me to ask questions*	*Parenting was factored into the service users’ recovery plan*	*Children’s needs were included in the service user’ recovery plan*
4. Assessment of vulnerable family member or carer’s needs	*The health service helped me with my needs as a carer and/or family member*	*The health service discussed the needs of the service users’ children.*	*The health service discussed the needs of the service users’ children.*
5. Provide or offer ongoing support to family and carers	*The health service helped me with what to do if the service user’s symptoms got worse*	*The health service provided parenting support to the service user.*	*The health service provided information about support available to children.*
6. Provide psycho-education to family and carers	*The health service talked to me about decisions impacting me as a carer and/or family member*	*The health service provided information/tips to the service user about parenting with a mental illness.*	*The health service helped children understand the service users’ illness.*
7. Provide or recommend referrals for family and carers	*The health service helped me to find support for my own needs*	*The health service offered the service user a referral for parenting support.*	*The health service offered referral for children to get support.*

## Data Availability

Not applicable.

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
