# Peer review of "Getting the FACS: A Protocol for Developing a Survey Instrument to Measure Carer and Family Engagement with Mental Health Services"

_ijerph, 2022, doi:10.3390/ijerph192316279_

Round 1

Reviewer 1 Report

I enjoyed reading your article and believe you establish a good case for the need for the protocol you suggest. That said,  here are some specific questions and suggestions for your consideration. First, you indicate that you establish a theoretical basis for your effort but Ia m not sure you do? At all events you treat the issue with an (internal) citation and little else? Perhaps a paragraph on the arguments for integrated/team care? Second, I suggest you spell out The National Institute for Health and Care Excellence when you refer to NICE guidelines on page 3.  Third, I was unclear how the "Northern Ireland and Australian practice audit tools" referenced on page 2 helped you develop your protocol/approach? Fourth, line 87 is in passive voice and I was therefore unable to determine its subject- the authors? Fourth, you refer at the outset of the article to children affected by the scenario that interests you and one could well imagine that mean to refer to youth who wind up caring for ill parents or taking on unusual responsibilities due to a parent or parent's illness. All good and well known. But then, later in your piece you allude specifically to adult children without explanation (line 191). I think you should be clear throughout on this point. Will you be asking adults to reflect on their experience as caregivers as adults or reflecting as adults on their experience as children or both?  Fifth, FN 53 refers to a reference group of experts who will review and suggest modifications to your instrument. I would have liked to know more about who comprises this group and why it is appropriate for this purpose? Last, Ia m skeptical about using social media to obtain your (convenience) samples (line 190 and ff.). Can you delimit a sample in a more targeted way? And why specifically would you not do so without the issue of self selection and unknown bias  resulting from a call on social media?

Author Response

Please see the attachment - it automatically gave it a file name different to what I had called it - but it includes our response to the review??  Not sure what is going on?  It is correct file but wrong name (not author cover letter).

Reviewer 2 Report

Thank you for giving me an opportunity to review this manuscript. It deals with the crucial issues regarding to measuring carer and family engagement with mental health services. This kind of surveys are crucial both for the health care system and family members. As the authors mentioned in the text: there are no valid or reliable measures of optimal ways that health care services might engage with family and carers. This gap limits the ability of health services to measure engagement practices for quality assurance, evaluation, and/or research purposes.

Seven ways that health services should engage with family and carers are correct without any doubts. Moreover, five stages proposed by the authors seem to be logic and well-prepared. The conctrucion of the study does not raise any concern as well.

My minor doubts are related with the sample. Therefore, describe please more precisely what stands behind the numbers (stage 2: n=12; stage 3: n=30&30; stage 4: n=250&250). What was the criterium? 

Using social media as a distributor sounds debatable. Possibility of non-random and not representative sample is on the horizon. What kind of sample it is? What about purposive or stratified sampling? Snowball sampling or Respondent Driven Sampling?

Write or describe a little more about the grey literature (line 134). It can allow the text to be more complete.

Additionally, some parts of the text were prepared untidly. Referance list has double numbers. There are missing numbers in Table 2. The title of the table should be rewrite without starting with the digit.

However, my general opinion about the study is positive.

Author Response

For some reason - with all my reviewer (1-4) uploads it changes the name of the file to being an author cover letter - it is our response to review.

Reviewer 3 Report

Congratulations to the authors on the theme. Very pertinent and necessary.

The theoretical framework is very well structured and deep in the relevant themes to the protocol.

I think it would be important to add a diagram with the stages planned in the protocol.

Also a chronogram with the forecast of the realization of each stage.

More than 40% of the references are more than 10 years old, see the possibility of using more recent references.

Author Response

For some reason it changes the name of my file to being a cover letter?? It is our response to review

Reviewer 4 Report

I think this paper should convert to an original study paper with Results section included. 

In your Method, you should include sample estimation for your construct validity and reliability. 

Construct validity such as exploratory factor analysis and/or confirmatory factor analysis should be included if possible. Detail of the analysis should be explained if it is a protocol paper. 

Researcher should also include testing the stability of the questionnaire by doing the test-retest reliability analysis. 

Comparison with other similar or more established or regarded as "gold standard" scale should be included to test the concurrent validity of the new questionnaire. 

Content validity should be included in the earlier phase of the questionnaire draft. 

Pre-testing of the questionnaire should be cognitive interview to ensure target participants understand and clear about the items in the questionnaire. 

Author Response

Somehow it automatically changes the name of the file to being a cover letter - it is not it is our review.

Round 2

Reviewer 4 Report

I think the paper should include the validation results, not just the methodology part. I leave the decision to the editor. Thank you. 
